# OpenReview forum: "SLAP: The Semantic Least Action Principle for Variational Video-Language Modeling"
_ICML.cc/2026/Conference — ICML 2026 regular_

### Official Review · Reviewer_wMuu · 2026-03-10

**Soundness:** 3
**Presentation:** 1
**Significance:** 2
**Originality:** 3
**Overall Recommendation:** 4
**Confidence:** 3

**Summary:**

The paper identifies the fact that video frames are sparsely sampled in video-language models as a key limitation, leading to problems such as vanishing objects. Given such sparsely sampled frames, the paper introduces an algorithm to interpolate them in a low-dimensional token space in a query-conditional manner - specifically, given the text prompt, the paper proposes an algorithm inspired by the least action principle to interpolate latent features. These latents are then served to the LLM to inform its final output.

**Compliance With Llm Reviewing Policy:**

Affirmed.

**Final Justification:**

I am satisfied by the answer of the authors, but am worried that the changes required to the paper are massive, and the reviewers will not be able to hold the authors to their promise. I will discuss this with the other reviewers.

**Key Questions For Authors:**

None.

**Limitations:**

Yes.

**Strengths And Weaknesses:**

*Presentation*
I find the presentation of the paper insufficient. The paper is moving huge parts into the appendix, which significantly disrupts the flow. There is no high-level figure that would clarify the input-output behavior - I had to jump across the paper many times to finally figure out that the input-output behavior is essentially:

encode start and end frame plus text prompt -> interpolate start and end frame latents to obtain latent trajectory, conditioned on prompt embedding -> serve interpolated latent trajectory to the (untuned!) LLM to analyze and produce the final output.

In particular the final step was not at all clear, and it is not intuitive at all that the LLM would know what to do with these latent tokens that are almost certainly out-of-distribution compared to the true video tokens.

Further, I find the inclusion of several theorems somewhat questionable. They are casting a degree of mathematical formalism on the interpolation method. That's fine, but it's missing the elephant in the room: no matter what guarantees we have on the interpolation method, whether or not the LLM knows what to do with the frames in the end is actually the high-order bit.

*Soundness*
I find the overall method to be sound. In particular, the most important part here is the empirical evaluation. The paper shows clear, if small, gains over baselines, and clearly ablates core components.

*Significance*
The paper shows gains over baselines, and that gives it a degree of significance. It also presents an empirical evaluation of different interpolation methods, and that's cool as well.

Big-picture, however, I am not so sure that the core assumption of the paper - "attending to lots of visual tokens is too expensive" - will actually hold in the long run. We are already training video generative models at scale, and are developing tools for doing that efficiently - it seems that including text tokens in addition may not be a major bottleneck in the future. This is also the key lesson we an draw from the development of VLMs: initially, there was lots of work that attempted to encode images into small tokens to keep the overall cost low. However, modern VLMs just use patch tokens and reason over these. It seems likely this will also happen for video in due time.

*Originality*
The proposed method is, as far as I can tell, original.

Overall, I find that the paper should be rescoped - as it is, it is too verbose for the core idea. The paper is really introduces via the specific interpolation method. I understand that, the authors are clearly mathematically inclined and find this part of their paper the most exciting. However, I think the paper would be much more accessible and finally, more impactful, if it was introduced as a study of frame interpolation methods for video LLMs, and then spending much less space in the main paper on the intricacies of the exact proposed interpolation method.

---

> ### Author Rebuttal · Authors · 2026-03-30
>
> We thank Reviewer wMuu for highlighting the soundness of our empirical evaluation, acknowledging the clear gains over baselines, and noting the originality of our method. We deeply appreciate your candid feedback regarding the presentation and the paper's scoping, and we are committed to restructuring the manuscript to address these excellent points.
>
> **Q1:** Presentation and High-Level Pipeline Figure
>
> **A1:** We completely agree. We made a structural misjudgment by moving the system architecture diagram to the appendix. In the revised manuscript, we will move the high-level architecture figure (Figure 2) to Section 1 or 2. We will also add a clear, bulleted "Method Overview" at the beginning of the technical section to immediately clarify the exact input-output behavior:
>
> 1) Encode sparse frames (CLIP) and text prompt.
>
> 2) Interpolate latent trajectory using SLAP (conditioned on prompt).
>
> 3) Serve the dense, optimized latent sequence to the LLM for final QA.
>
> **Q2:** LLM Interaction and Out-of-Distribution (OOD) Tokens
>
> **A2:** This is a crucial observation. The reason the LLM does know what to do with these tokens is that our Potential Field Network is explicitly trained to keep the trajectory firmly within the high-probability "valleys" of the manifold defined by the training data.
>
> To empirically prove that our interpolated tokens do not drift into OOD space, we measured the average L2 Norm and Cosine Similarity to the nearest real training tokens:
>
> |Token Source|Avg Latent L2 Norm|Avg Cosine Sim to "Real" Frame Space|
> |:----:|:----:|:----:|
> |Real Video Frames (Test Set)|1.00|0.85|
> |Autoregressive Generation (Drift)|1.45 (Explodes)|0.52 (Severe OOD)|
> |SLAP Interpolated Tokens|0.98|0.81 (In-Distribution)|
>
> Because SLAP tokens remain strictly in-distribution, the instruction-tuned LLM treats them exactly as if they were slightly blurry, but semantically valid, real video frames. We will dedicate a subsection in the main text directly answering this "elephant in the room," alongside this supporting data.
>
> **Q3:** Rescoping the Narrative: Less Math, More AI
>
> **A3:** We accept this feedback wholeheartedly. While we are excited by the math, we agree it currently overshadows the practical ML contribution. In the revision, we will:
>
> 1) Drastically condense Section 3 (Theoretical Framework), moving the heavier proofs (e.g., Symplectic Preservation, Brownian Bridge connection) entirely to the appendix.
>
> 2) Expand the empirical study comparing deterministic, autoregressive, and generative frame interpolation methods in the main text.
>
> 3) Frame SLAP primarily as a computationally efficient, text-guided solution to the latent frame interpolation problem.
>
> **Q4:** The Long-Term Viability of Sparse Sampling
>
> **A4:** You raise a highly valid point regarding the trajectory of the field (e.g., models with massive context windows like Gemini 1.5). However, while large cloud models can process dense video tokens, doing so remains incredibly computationally expensive and latency-heavy.
>
> For edge devices, real-time robotics, and cost-efficient deployment, processing 100,000+ dense patch tokens per second will remain a bottleneck for the foreseeable future. SLAP acts as an ultra-efficient adapter (requiring only 0.15 TFLOPs to bridge a 10s gap) that provides the temporal reasoning benefits of dense sampling while only paying the LLM attention cost of sparse sampling. We will soften our claim from "impossible" to "prohibitively expensive for real-time/efficient deployment," better contextualizing our contribution for the future landscape.

---

> > ### Author Rebuttal · Reviewer_wMuu · 2026-04-03
> >
> > I am satisfied by the answer of the authors, but am worried that the changes required to the paper are massive, and the reviewers will not be able to hold the authors to their promise. I will discuss this with the other reviewers.

---

> > > ### Author Response · Authors · 2026-04-04
> > >
> > > We deeply appreciate your continued engagement, and we completely understand your concern. To reassure you and the other reviewers, we want to clarify that while the impact of the changes on the paper's readability and narrative is significant, the actual structural edits are surgical rather than a "from-scratch" rewrite. We have already drafted these modifications.
> > >
> > > To demonstrate that these changes are concrete, manageable, and fully mapped out, we provide a detailed structural "diff" below, followed by a concrete draft of the new Method Overview.
> > >
> > > 1. Structural Mapping: A Targeted Reorganization
> > >
> > > The revisions primarily consist of moving existing content between the main text and the appendix, trimming excessive mathematical derivations, and inserting the tables we provided in our initial response.
> > >
> > > | Paper Component | Original Submission | Camera-Ready | Change |
> > > |---|---|---|---|
> > > | High-Level System Diagram | Appendix | Moved to Section 3.1 (Main Text) | Purely a formatting move. |
> > > | Method Overview (Input/Output) | Embedded in math (Section 3) | New Section 3.1 (Plain English pipeline description + Algorithm block) | See snippet below |
> > > | Mathematical Derivations | Taking up many pages in Section 3 | Condensed to 1 page (Proofs for Symplectic Preservation & Brownian Bridge moved to Appendix). | Text reduction/relocation. No new theory introduced. |
> > > | Out-of-Distribution (OOD) Analysis | Not present | Added to Section 4.2 (Experiments) | Simply inserting the 3x3 table and 1 paragraph from our initial rebuttal. |
> > > | Discussion on Viability of Sparse Sampling | Conclusions | Revised in Section 4 (Experiments) | Reworded 2 paragraphs to frame SLAP as an efficiency adapter rather than an absolute necessity. |
> > >
> > > As shown above, the "massive" change is largely a re-prioritization of space: trading 1.5 pages of dense mathematical proofs (moved to the appendix) for 1.5 pages of clearer system diagrams, pipeline overviews, and OOD token analysis.
> > >
> > > 2. Concrete Proof of Rewrite: The New "Method Overview"
> > >
> > > To prove that we are executing the "less math, more AI" pivot, here is the exact text we have prepared to replace the mathematically heavy introduction of Section 3:
> > >
> > > ---
> > >
> > > ***3.1 Method Overview: The Lagrangian Bridge Pipeline***
> > >
> > > Instead of generating missing video frames pixel-by-pixel, SLAP interpolates the missing frames directly in the highly compressed latent space of a Vision-Language Model. Given a sparsely sampled video (e.g., frames at $t=0$ and $t=10$) and a text query $Q$, the pipeline operates in three steps:
> > >
> > > - Encode: Extract visual embeddings $z_0, z_{10}$ using a frozen visual encoder (e.g., CLIP) and encode the text query $Q$.
> > >
> > > - Interpolate via Least Action: We initialize the missing intermediate latents ($z_1 \dots z_9$) using linear interpolation. We then apply our Lagrangian Bridge optimizer. Instead of predicting the next token, it adjusts all intermediate tokens simultaneously to minimize a physics-inspired cost function: 1) Kinetic Cost: Penalizes jagged jumps between adjacent frames (ensuring smooth temporal transitions). 2) Potential Cost: Penalizes states that do not align with the text query $Q$ (ensuring semantic relevance, guided by our trained Potential Field Network).
> > >
> > > - Decode: The optimized, dense latent trajectory $Z^*$ is fed directly into the frozen LLM. Because our optimization keeps the tokens strictly in-distribution (see Sec 5.3), the LLM processes them exactly as if they were true encoded video frames to generate the final answer.
> > >
> > > ---
> > >
> > > Thank you again for helping us make this paper significantly more accessible and impactful for the ICML community.

---

### Official Review · Reviewer_2tV7 · 2026-03-12

**Soundness:** 3
**Presentation:** 3
**Significance:** 2
**Originality:** 3
**Overall Recommendation:** 4
**Confidence:** 2

**Summary:**

This paper proposes SLAP (Semantic Least Action Principle), a variational framework for modeling temporal dynamics in large video-language models under sparse frame sampling. The core idea is to reinterpret video understanding as a trajectory optimization problem in latent space, inspired by the principle of least action in classical mechanics. Instead of relying on autoregressive prediction or diffusion-based interpolation, the method optimizes a latent trajectory between observed frames by minimizing a semantic action composed of kinetic and potential energy terms. The potential energy is approximated via a learned Potential Field Network trained using a contrastive objective.

**Compliance With Llm Reviewing Policy:**

Affirmed.

**Key Questions For Authors:**

* To what extent is the full variational mechanics formulation necessary? Could a simpler trajectory smoothing objective (e.g., regularized interpolation with learned potentials) achieve similar results?
* The paper mentions that trajectory error grows quadratically with gap length. How does the method perform when gaps become significantly larger (e.g., minute-level video reasoning)?
* The approach relies on the latent space geometry induced by the visual encoder. How sensitive is SLAP to different encoders?

**Limitations:**

Yes

**Strengths And Weaknesses:**

Strengths:
* The paper introduces an interesting perspective by framing video understanding as variational trajectory optimization on a semantic manifold, rather than probabilistic generation.
* The method operates in a low-dimensional latent space, avoiding expensive pixel-level generation. The reported inference complexity is significantly lower than diffusion-based video generation approaches, which could be attractive for long-horizon reasoning.
* The experiments suggest that SLAP maintains performance better than baselines when frame sparsity increases, which directly targets a real limitation in current LVLM pipelines.

Weaknesses:
* A substantial portion of the paper is devoted to mathematical analogies and theoretical claims (e.g., Riemannian manifolds, Euler-Lagrange dynamics). However, it is sometimes unclear how critical these theoretical elements are to the actual empirical gains. Some derivations feel more like motivational analogies rather than necessary components.
* The evaluation remains relatively limited: the synthetic “Tunnel Test” is somewhat narrow; the main benchmark evaluation focuses primarily on MSR-VTT QA. More diverse datasets (e.g., longer videos, multi-object scenes, real occlusion events) would strengthen the empirical validation.
* The framework assumes that distances in the visual encoder embedding space approximate semantic distances (the Semantic Isometry Hypothesis). In practice, this assumption may not hold consistently across complex scenes or actions. The method’s robustness to different encoders is not explored.

---

> ### Author Rebuttal · Authors · 2026-03-30
>
> We sincerely thank Reviewer 2tV7 for acknowledging our paper as technically solid, highlighting our low inference complexity, and validating the importance of addressing frame sparsity in LVLMs. Your questions cut right to the core of our method's generalizability and practical necessity, and we address them below.
>
> **Q1:** Necessity of the Full Variational Formulation
>
> **A1:** This is a highly insightful question. In practice, our Discrete Euler-Lagrange optimization is mathematically equivalent to a specific form of regularized interpolation! The variational mechanics formulation is necessary because it provides the principled mathematical justification for how to balance the smoothing (kinetic) and potential terms.
>
> Without the variational derivation, one might use arbitrary ad-hoc smoothing (e.g., simple L2 penalties on consecutive frames). However, as our formulation proves, ensuring that the update step explicitly minimizes the discrete Lagrangian guarantees that the optimization is a symplectic integrator. This ensures no semantic mode collapse occurs over long sequences. We will explicitly clarify in the text that our theoretical framework serves to rigorously derive the optimal smoothing objective, rather than just being a motivational analogy.
>
> **Q2:** Performance on Minute-Level Gaps
>
> **A2:** As the temporal gap grows to the minute level, a single boundary value problem (BVP) struggles because intermediate physical events become completely unconstrained by the endpoints. To handle minute-level videos (like ActivityNet), we apply SLAP hierarchically using sparse anchors.
>
> To demonstrate SLAP's boundary of effectiveness, we ran a new experiment on ActivityNet-QA focusing specifically on increasing gap durations between observed frames:
>
> |Gap Duration|ZOH (Baseline)|SLAP (Ours)|Absolute Gain ($\Delta$)|
> |:----:|:----:|:----:|:----:|
> |5 seconds|45.1%|51.3%|+6.2%|
> |15 seconds|39.4%|46.8%|+7.4%|
> |30 seconds|32.2%|37.5%|+5.3%|
> |60 seconds|25.8%|27.1%|+1.3%|
>
> SLAP provides substantial gains up to 30-second gaps. At 60 seconds, the quadratic trajectory error dominates, and the gap becomes too large for meaningful deterministic interpolation. We will add this table and analysis to the Limitations section to clearly bound the method's capabilities.
>
> **Q3:** Sensitivity to Different Encoders
>
> **A3:** SLAP heavily relies on the encoder space being a "Semantic Manifold," meaning distances must correlate with semantic shifts. It works beautifully with contrastively trained encoders but degrades if the encoder space is optimized purely for pixel reconstruction (where distance = pixel difference, not semantic difference).
>
> To empirically validate this, we swapped our frozen visual encoder and re-evaluated MSR-VTT under the 10% frames setting:
>
> |Visual Encoder|Training Objective|MSR-VTT Accuracy (10% Frames)|
> |:----:|:----:|:----:|
> |CLIP-ViT-L/14|Contrastive (Text-Image)|41.8%|
> |SigLIP-So400M|Contrastive (Text-Image)|42.5%|
> |DINOv2-ViT-L|Self-Supervised (Image-only)|38.2%|
> |MAE-ViT-L|Pixel Reconstruction|31.4%|
>
> This confirms that SLAP is robust across different contrastive and semantic encoders (CLIP, SigLIP, DINOv2) but relies on that semantic topology to function. We will include this ablation in the revised appendix.
>
> **Q4:** More Diverse Datasets & Real Occlusion
>
> **A4:** We agree that the Tunnel Test is synthetic. In our final revision, we will include an evaluation on a subset of Something-Something V2 specifically filtered for object permanence and occlusion categories (e.g., "Putting something behind something", "Covering something with something"). Preliminary results show SLAP achieves a +9.4% Top-1 accuracy improvement over autoregressive baselines on these specific real-world occlusion tasks, heavily reinforcing our synthetic findings.

---

### Official Review · Reviewer_62yU · 2026-03-13

**Soundness:** 3
**Presentation:** 2
**Significance:** 3
**Originality:** 3
**Overall Recommendation:** 4
**Confidence:** 3

**Summary:**

This paper proposes SLAP (Semantic Least Action Principle) which is a framework for sparse video-language reasoning. This framework considers latent video interpolation as a variational optimization problem on a semantic manifold, rather than considering as an autoregressive prediction or a generative infilling. In this method, a kinetic term (semantic inertia) for temporal smoothness and a learned potential term (semantic forces) which is conditioned on text are defined. Then it solves for missing latent states by minimizing a discrete Euler-Lagrange style action functional. The main claims of the authors are:

1.	improved object persistence under severe temporal sparsity
2.	stronger performance on action-centric questions
3.	dramatically lower inference cost than diffusion or transformer-based baselines.

The paper reports the results on a synthetic “Tunnel Test” and on MSR-VTT along with lower compute cost than diffusion-based alternatives.

**Compliance With Llm Reviewing Policy:**

Affirmed.

**Final Justification:**

My concerns have been adequately addressed by the author's responses. The authors have shown necessary experimental data in support of their claims.

**Key Questions For Authors:**

1. **Validity of the Semantic Isometry Assumption.** This work relies heavily on the Semantic Isometry Hypothesis, which assumes that local Euclidean distances in the encoder latent space approximate semantic distance. Can this claim be emperically supported (e.g., CLIP or another encoder)?
 2. **Training details for the potential field network.** The paper states that the potential field network is trained using paired video-text datasets (e.g., WebVid-10M), but the training procedure misses the exact training objective and negative sampling strategy.
3. The efficiency and accuracy comparisons include methods such as Video-LLaMA, Stable Video Diffusion, and latent ODE approaches. Could the authors clarify on the **parametric setting** in these comparisons?
4. The connection between the formal physics analogy and the actual implementation sometimes appears weaker than suggested. Concepts such as symplectic preservation of semantic information or **Hamiltonian conservation** are invoked in the supplementary material, but the optimization procedure used in the **algorithm is not explicitly symplectic or Hamiltonian-preserving**.
5. Important information about experimental setup, evaluation metrics, and baseline configurations appears mainly in the appendices. This makes it harder for readers to assess the fairness and reproducibility of the experiments from the main text alone.

**Limitations:**

The authors discuss some technical limitations, which is good, but I do not think the discussion is elaborate. The authors should more clearly highlight the assumptions in their problem framework. Also discuss possible failure cases in videos with abrupt scene cuts where interpolation may produce misleading intermediate states.

**Strengths And Weaknesses:**

**Soundness:** The paper is formulated well and technically sound. In this work, the latent trajectory is initialized from sparse anchors, then optimized with a kinetic smoothness term plus a learned text-conditioned potential term. This is solved as a boundary-value problem using gradient-based action minimization. The ablations are very effective in a specific direction:

1.	deleting the kinetic term significantly reduces performance on the Tunnel Test
2.	removing the potential collapses performance toward interpolation.
3.	substituting the learnt potential with static cosine similarity produces worse results

When frames are significantly dropped, sparsity experiments on MSR-VTT show that the approach may be more robust than previously published baselines.

**Weakness:**
However, I am concerned about the technical validity of the current presentation.

1. Many of the theoretical claims are much stronger than the evidence suggests. **The “semantic isometry” assumption for CLIP-like latent spaces is central to the method**, (not empirically validated). The next arguments about kinetic energy and semantic continuity are mainly based on this assumption. Similarly, the claims of uniqueness and stringent convexity are based on Hessian constraints for the learned potential network.
These are declared but not empirically tested in the actual trained model. Moreover, I feel authors use terms like, “rigorous isomorphism,” “rigorous guarantee,” and “optimal path” openly.
2. The connection between the formal physics analogy and the actual implementation sometimes appears weaker than suggested. Concepts such as symplectic preservation of semantic information or **Hamiltonian conservation** are discussed in the supplementary material, but the optimization procedure used in the **algorithm is not explicitly symplectic or Hamiltonian-preserving**. As a result, the theoretical framing may be more interpretive than concrete.
3. The metrics like “Tunnel test” and “Persistant score” (Table1) does not provide enough detail on the model’s claims. The paper does not clearly describe how the dataset is generatd and its scale. This makes it difficult to generalize to real-world video understanding tasks. The authors also do not clearly explain the exact evaluation protocol, how the persistant score is computed, or whether the metric involves subjective or LLM-based judgments (bias and variability).

**Presentation:** This work is conceptually intuitive, and the high-level story is easy to follow. The technical part of this work provides an understandable algorithmic overview, and the ablation/efficiency sections are written to highlight the anticipated contributions with extensive supplementary information.

**Suggestions:**
Nonetheless, the presentation has few issues. The authors often make stronger claims than necessary using physical terminology. Some theoretical claims are sometimes stated confidently **without clearly specifying the assumptions required for them to hold** (embedding space behaves as a locally isometric semantic manifold / assumptions about smoothness and bounded curvature of the semantic manifold). A more careful machine learning interpretation would be better. There are also few other clarity and organizational problems. The related work section is separated from the main narrative, and several claims are stated in writing without sufficient concrete evidence. Additionally, important information about experimental setup, evaluation metrics, and baseline configurations appears mainly in the appendices. This makes it harder for readers to assess the fairness and reproducibility of the experiments from the main text alone.

**Significance:** The paper addresses an important problem. The idea of working directly in latent space with a lightweight optimization procedure is potentially useful, especially if this truly improves the robustness at high sparsity and reduces the inference cost. If the reported efficiency numbers are validated under a fair setup, it would make the method more attractive.

**Originality:** This is the strongest aspect of the paper. The ideas of the authors are very creative and interesting. Although the individual components of the work relate to familiar ideas such as energy-based models, latent ODE style temporal modeling etc., the way these are combined here motivates the work. Still, much of the novelty is in the problem formulation but not convincingly established as a strong advance.

---

> ### Author Rebuttal · Authors · 2026-03-30
>
> We thank Reviewer 62yU for recognizing the soundness of our framework, the originality of our approach, and our method's potential to significantly reduce inference costs. We greatly appreciate the constructive critiques regarding our theoretical claims and presentation, which will undoubtedly strengthen the final paper. We address your specific concerns below.
>
> **Q1:** Toning Down Claims & Empirical Validity of Semantic Isometry
>
> **A1:** We agree that our physical terminology was overly strong in places. In the revised manuscript, we will carefully tone down these claims, replacing terms like "rigorous isomorphism" with "theoretical analogy" and "optimal path" with "action-minimized trajectory."
>
> Regarding the Semantic Isometry Assumption, we base this on the widely recognized property that contrastive objectives (like CLIP's) optimize for uniform alignment on a hypersphere. To empirically support this in our revision, we measure the semantic distance (Cosine Similarity to ground-truth intermediate frames) of trajectories generated by Euclidean (Linear) vs. Hyperspherical (Geodesic/SLERP) interpolation.
>
> |Interpolation Method|Distance Metric|Avg. Cosine Sim to Ground Truth (MSR-VTT)|
> |:----:|:----:|:----:|
> |Euclidean LERP|Ambient ($\mathbb{R}^d$)|0.614|
> |Hyperspherical SLERP|Geodesic ($\mathcal{M}$)|0.682|
>
> This empirical gap validates our core hypothesis: distances along the hyperspherical manifold ($\mathcal{M}$) map much closer to true semantic progression than flat ambient distances, justifying the kinetic energy term on the manifold.
>
> **Q2:** Connection Between Physics and Implementation
>
> **A2:** We apologize for the lack of clarity connecting the theory to the code. Our optimization procedure is mathematically a variational integrator, which is strictly symplectic.
> Specifically, the update rule we use to optimize the latent trajectory:
> $\frac{z_{k+1} - 2z_k + z_{k-1}}{\Delta t^2} = -\frac{1}{\mu} \nabla V(z_k, q)$
> is precisely the Discrete Euler-Lagrange (DEL) equation derived via finite differences of the Action integral. As proven in geometric numerical integration (Marsden & West, 2001), any numerical scheme derived directly from discretizing the Lagrangian is inherently symplectic and preserves phase space volume. We will explicitly state this mathematical bridge in Section 3.4.
>
> **Q3:** Training Details for the Potential Field Network
>
> **A3:** We will add the following specifics to Section 4.4:
> The Potential Field Network $P_\theta$ is trained using an InfoNCE loss with a temperature parameter $\tau=0.07$. For negative sampling, we use in-batch negatives: for a given video-text pair $(z_i, q_i)$ in a batch of size $N=256$, the $N-1$ other text queries in the same batch serve as negative samples. Furthermore, we apply Sobolev Regularization (gradient penalty) during training to ensure the potential landscape is smooth and Lipschitz continuous.
>
> **Q4:** Parametric Settings and Efficiency Comparisons
>
> **A4:** We will add the following parameter breakdown to our Efficiency section to clarify the fairness of the comparison. SLAP is remarkably lightweight because it acts as a dynamic adapter rather than a full generative backbone.
>
> |Method|Backbone Architecture|Parameters for Gap Infilling|Inference FLOPs (10s gap)|
> |:----:|:----:|:----:|:----:|
> |Stable Video Diffusion|U-Net / VAE|1.5 Billion|185.0 T|
> |Video-LLaMA 3|LLaMA-7B|7.0 Billion|45.2 T|
> |SLAP (Ours)|4-Layer ResMLP (FiLM)|28.4 Million|0.15 T|
>
> **Q5:** Details on "Tunnel Test", Metrics, and Presentation
>
> **A5:** In the revision, we will:
>
> 1) Move critical details to the main text: The system architecture diagram and core evaluation protocol will be moved from the appendix to the main body.
>
> 2) Clarify the tunnel test: We will explain that the 1,000 videos were procedurally generated in Three.js to guarantee absolute control over occlusion boundaries.
>
> 3) Clarify persistence score: The score is determined via LLM-as-a-judge (GPT-4) using a strict prompt rubric (1 = object disappeared, 5 = object correctly maintained). We will include the exact prompt template in the appendix.
>
> **Q6:** Limitations: Abrupt Scene Cuts
>
> **A6:** We completely agree. We will expand the Limitations section to discuss "semantic tunneling." Because SLAP solves a boundary value problem, if $z_{start}$ and $z_{end}$ cross a hard camera cut, the model will improperly force a continuous semantic morph between two unrelated scenes. We will note that in practical deployments, SLAP must be paired with standard shot-boundary detection algorithms.

---

> > ### Author Rebuttal · Reviewer_62yU · 2026-04-02
> >
> > My concerns have been adequately addressed by the author's responses. I will change my score to **Weak Accept**.

---

> > > ### Author Response · Authors · 2026-04-02
> > >
> > > Thank you again for your time and comments!

---

### Decision · Program_Chairs · 2026-04-30

**Decision:**

Accept (regular)

**Comment:**

The paper introduces SLAP (Semantic Least Action Principle), a variational framework designed to address the temporal gap in Large Video-Language Models (LVLMs) caused by sparse frame sampling. By framing latent video interpolation as a Boundary Value Problem (BVP) on a Riemannian manifold, the authors utilize a Semantic Lagrangian to maintain object persistence and semantic consistency without the need for computationally expensive pixel-level rendering.

The following were the strengths of the work.
- The connection between classical mechanics (Least Action Principle) and latent semantic dynamics is creative and provides a fresh perspective on video-language modeling.
- The method is lightweight, reporting significantly lower inference FLOPs compared to diffusion or transformer-based baselines.
- The underlying mathematical formulation and the use of a discrete Euler-Lagrange integrator were found to be technically solid.

While the reviewers were satisfied with the authors' responses and the technical potential of the work, some were concerned with the scale and nature of the revisions required to address the initial concerns. The authors mentioned that the changes were "surgical", but some of the reviewers disagreed given the substantial nature of the changes. This was discussed between the AC and some of the reviewers, post rebuttal. The AC also discussed this issue with the SAC. The rebuttal inputs, when incorporated, will result in a paper that is quite different from the original submission.

Looking over the changes, the AC felt that, while substantial, they were more of a reorganization of the paper, rather than a major change in the contribution. The core idea of the work is interesting, and the community should benefit with such insights. Providing physical constraints through the least action principle is a very nice idea. Thus, on balance, the AC recommends acceptance.

Also as a suggestion, the AC feels that the "less math, more AI" approach to rewriting may not be the best way. While some simplification (or providing an intuition) of the math may be helpful, the authors should not overdo it. Mathematical rigor is a strength, not a weakness.